# Drying of the Natural Fibers as A Solvent-Free Way to Improve the Cellulose-Filled Polymer Composite Performance

**DOI:** 10.3390/polym12020484

**Published:** 2020-02-21

**Authors:** Stefan Cichosz, Anna Masek

**Affiliations:** Institute of Polymer and Dye Technology, Faculty of Chemistry, Lodz University of Technology, Stefanowskiego 12/16, 90-924 Lodz, Poland; stefan.cichosz@dokt.p.lodz.pl

**Keywords:** cellulose fibers, moisture content, ethylene–norbornene copolymer

## Abstract

When considering cellulose (UFC100) modification, most of the processes employ various solvents in the role of the reaction environment. The following article addresses a solvent-free method, thermal drying, which causes a moisture content decrease in cellulose fibers. Herein, the moisture content in UFC100 was analyzed with spectroscopic methods, thermogravimetric analysis, and differential scanning calorimetry. During water desorption, a moisture content drop from approximately 6% to 1% was evidenced. Moreover, drying may bring about a specific variation in cellulose’s chemical structure. These changes affected the cellulose-filled polymer composite’s properties, e.g., an increase in tensile strength from 17 MPa for the not-dried UFC100 to approximately 30 MPa (dried cellulose; 24 h, 100 °C) was observed. Furthermore, the obtained tensile test results were in good correspondence with Payne effect values, which changed from 0.82 MPa (not-dried UFC100) to 1.21 MPa (dried fibers). This raise proves the reinforcing nature of dried UFC100, as the Payne effect is dependent on the filler structure’s development within a polymer matrix. This finding paves new opportunities for natural fiber applications in polymer composites by enabling a solvent-free and efficient cellulose modification approach that fulfils the sustainable development rules.

## 1. Introduction

Sustainable development rules are of the highest importance when considering reliable resource management [1]. Regarding the plastic industry, nowadays, a great challenge is to obtain lower amounts of plastic waste [2] and to minimize environmental pollution [3]. Fortunately, the development of modern chemistry has enabled us to transform our natural resources in order to create new matter from existing ones to benefit society [4,5,6,7]. This has greatly enriched modern living and increased the quality of life [8].

As has been mentioned, raising environmental awareness has put stress on developing materials that are less harmful to natural habitats. In the following research, an application of fully biodegradable natural fibers in the role of polymer composite fillers is presented. However, highly polar cellulose exhibits a poor adhesion to hydrophobic polymer matrixes [9,10,11], and, therefore, low polymer composite performance has been observed in this context. Consequently, cellulose requires a surface modification process to be carried out. Nonetheless, most treatments involve solvent employment as a reaction media, which is not good for the environment.

Solvents contribute to the increased toxicity, hazard, pollution, and rise of waste treatment issues. Moreover, they account for a major source of the wasted mass of a given process or a synthetic pathway. On the other hand, solvents are essential when considering the mass either heat transfer facilitation. Moreover, an appropriate solvent choice may contribute to the obtainment of the proper reaction rate, selectivity, or position of the chemical equilibrium [12].

Luckily, many common solvents, e.g., acetone [13], methanol [14], ethanol [15], butanol [16], isopropanol [17], ethyl acetate [18], and 2-methyltetrahydrofuran [19], may be produced from renewable feedstocks rather than from petroleum [20]. Additionally, the list of preferably used solvents [20,21] includes water [22], acetone [23], ethanol [24], 2-propanol [25], 1-propanol [26], methanol [27], 1-butanol [28]. Some repetitions between these two lists may be visible.

What should be emphasized is that water (in the role of a solvent) has drawn a lot of attention in recent years [29,30,31,32]. It is cheap, available, non-toxic, and non-flammable. Water seems to be attractive from both the economic and environmental points of view. However, it has been found that water is a poor solvent when carrying out organic transformations, as it exhibits a low solubility of organic compounds. Therefore, water is considered a contaminant most of the time [12]. However, it has been proven that, thanks to its unique structure and physicochemical properties, water might lead to particular interactions like polarity [33], hydrogen bonding [34], hydrophobic effects [35], and trans-phase interactions [36]; therefore, it determines the reaction course [12].

Furthermore, regarding appropriate solvent choice, not only the solvent’s toxicity should be taken into consideration; atom efficiency [37], reaction yield [38], workup [39] and purification process parameters [40] also play important roles. Nonetheless, water enables a broad scope of reactions and, undoubtedly, may lead to additional sustainability benefits that enhance the overall environmental impact of a given process [12].

It has been found that cellulose requires some specific surface treatments in order to obtain a polymer composite of a sufficient performance. Fortunately, there are many known examples of chemical modifications that are carried out in neat conditions and in solvent-free environments, e.g., acetylation [41,42], carboxylation [43,44], and fluorination [45]. Moreover, cellulose is not only being modified in mild conditions; other treatments include lignocellulose acetylation [46] and lignin acetylation [47].

Another interesting process of cellulose performance improvement is hornification, which relies on the carrying out of wetting–drying cycles of the biopolymer [48]. Subsequently, some changes in the chemical structure of the natural fiber occur [49,50]. This allows for a greater dimensional stability and a lower degradation via increases in molecular packing [51,52]. Hornification is commonly carried out in a water environment, and the whole process may be controlled by, e.g., cycle amount and drying parameters [51,53].

The effect of water amount on polymer composite properties has been researched [54,55,56]. However, this effect has never been investigated deeply enough. The origins of different properties have not been explained. This research provides the reader with a wide range of new and precise data that considers the properties of both cellulose and polymer composites regarding different biofiller moisture contents (controlled via a drying time). On the basis of the gathered information, it could be observed that with a simple drying process of cellulose, polymer composite performance may be doubled. This article is an introduction to research analysis on the consequences of water contents in cellulose on the properties of natural fiber-loaded polymer composite properties, as well as biopolymer structures and features.

## 2. Materials and Methods

### 2.1. Materials

The Arbocel^®^ UFC100 Ultrafine Cellulose for Paper and Board Coating from J. Rettenmaier & Soehne (Rosenberg, Germany) was the type of cellulose that was used in this research. It is in a powder form (white, odorless), and its density is about 1.3 g/cm^3^. It is insoluble in water and fats. Nevertheless, this material exhibits a high water-binding capacity (even at high temperatures and shearing forces). Its average fiber length is 8–14 μm.

As a polymer matrix thermoplastic elastomer, the ethylene–norbornene copolymer (TOPAS^®^ Elastomer E-140 from TOPAS Advanced Polymers^®^, Raunheim, Germany) was employed. This material is a high-performance alternative to traditional flexible materials for use in a broad range of applications such as medical devices, injection molding articles for the optical industry, and other uses in the packaging industry. From the processing point of view, the crucial aspects are the melting temperature, which has reported to be 84 °C, and the Vicat softening temperature, which has been determined to be 64 °C. The bulk density of the material is in between 450 and 550 g/dm^3^. Figure 1 reveals the structure of the ethylene–norbornene composite.

### 2.2. Cellulose Fibers Samples of Varied Moisture Contents Preparation

One of the research aims was to examine the influence of moisture content in cellulose fibers on polymer composite properties. Therefore, UFC100 was dried for various times in order to obtain different water contents in such a prepared filler. Drying was performed in a Binder^®^ oven at 100 °C for the drying times given in Table 1. Cellulose fibers were evenly distributed in the crystallizer (70 × 40 mm).

### 2.3. Polymer Composite Samples Preparation

Cellulose fibers were dried for 24 h at 100 °C (Binder^®^ oven; crystallizer 70 × 40 mm) before being incorporated into the TOPAS. Then, the polymer matrix (86 wt.%) and cellulose fibers (14 wt%) were mixed in a micromixer (Brabender Lab-Station from Plasti-Corder with Julabo cooling system) at 110 °C for 30 min (50 rpm). Next, such prepared materials were put between two rolling mills prior to orienting the fibers. The prepared mixture was plasticized at 100 °C for 30 min in an oven and then put between two roll mills with 100 × 200 mm rolls at a roll temperature of 20–25 °C and a friction of 1:1.1 for approximately 45 s. The last step was to compress the composite plates between two steel molds and Teflon sheets, in a hydraulic press at 160 °C (electrically heated platens) for 10 min at approximately 125 bar.

### 2.4. Cellulose Fibers Properties Characterization

#### 2.4.1. Fourier Transform Infrared Spectroscopy (FT-IR)

Fourier transform infrared spectroscopy (FT-IR) absorbance spectra were investigated within the 4000–400 cm^−1^ range (64 scans, adsorption mode). The experiment was performed with the use of a Thermo Scientific Nicolet 6700 FT-IR spectrometer that was equipped with diamond Smart Orbit ATR sampling accessory. Cellulose fibers were dried for 24 h at 100 °C (Binder^®^ oven; crystallizer 70 × 40 mm) before being analyzed.

#### 2.4.2. Thermogravimetric Analysis (TGA)

Thermogravimetric analysis (TGA) was used in order to get acquainted with the thermal degradation process of cellulose fibers by detecting the mass loss as a function of raising the temperature in the range from 25–600 °C (heating rate: 10 °C/min; Ar 60 cm^3^/min). A Mettler Toledo TGA/DSC 1 STARe System equipped with a GC10 Gas Controller was employed in this investigation. Activation energy (E_A_) values were calculated with the use of Broido’s method [57]:(1)$$y=\frac{m_{t}-m_{\infty }}{m_{0}-m_{\infty }}$$
(2)$$\ln\left[\ln\left(\frac{1}{y}\right)\right]=-\frac{E_{A}}{R}\cdot \frac{1}{T}+C\;as\;a\;linear\;function:Y=aX+b$$
(3)$$where:Y=\ln\left[\ln\left(\frac{1}{y}\right)\right],\;X=\frac{1}{T},a=-\frac{E_{A}}{R},b=C$$
(4)$$therefore\;E_{A}=-a\cdot R$$
where:$m_{t}$—specimen mass at the time *t* [g];$m_{0}$—specimen mass at the beginning of considered decomposition step [g];$m_{\infty }$—specimen mass at the end of considered decomposition step [g];T—temperature [K];R—gas constant [8.31 J/(mol·K)].

Cellulose fibers were dried for 24 h at 100 °C (Binder^®^ oven; crystallizer 70 × 40 mm) before being investigated (apart from the analysis of the fibers of different moisture contents).

#### 2.4.3. Fischer Titration

Fischer titration was carried out with the use of Hydranal Solvent E and Hydranal Titrant 5E, which were supplied by Fluka^®^. For each experiment, approximately 1.5 g of cellulose samples and 50 mL of Hydranal Solvent E were taken. The water content in the natural filler was established as follows [58]:(5)$$BI_{2}+BSO_{2}+B+H_{2}O\to 2BH^{+}I^{-}+BSO_{3}$$
(6)$$BSO_{3}+ROH\to BH^{+}ROSO_{3}{}^{-}$$

The anode solution consisted of an alcohol (ROH), a base (B), SO_2_, and I_2_. The Pt anode generated I_2_ when current was provided through the electric circuit. The net reaction, as shown above, was the oxidation of SO_2_ by I_2_. One mole of I_2_ was consumed for each mole of H_2_O. In other words, 2 moles of electrons were consumed per mole of water.

The measurement was done with the use of a TitroLine Alpha (Schott^®^) device (Mainz, Germany). Cellulose fibers were dried for 24 h at 100 °C (Binder^®^ oven; crystallizer 70 × 40 mm) and then left in ambient conditions for 45 min before being investigated (it was impossible to carry out an accurate measurement immediately after taking the sample out of a dryer—too high a water adsorption rate).

#### 2.4.4. Detection of Mass Change during Moisture Adsorption/Desorption

The moisture desorption of the cellulose fibers was tested by putting a cellulose sample of approximately 1.5 g in a dryer (100 °C) in a small weighing bottle. Its mass was examined over 24 h with the use of analytical weight (Radwag^®^). Measurements were done each 15 min during the first 90 min, and then the interval time was increased to 30 min until 360th minute of the experiment. The last measurement was done after 1440 min (24 h) of the drying process.

The moisture adsorption of the cellulose fibers was tested by putting a cellulose sample of approximately 1.5 g in a small weighing bottle, drying it for 24 h at 100 °C into a dryer, taking it out after the given time, and leaving in an ambient condition. Again, its mass was examined over 24 h with the use of analytical weight. The intervals between the measurements were the same as before. The sample mass values were transformed into the normalized moisture levels (Y) according to the following equations:(7)$$Y=\{\begin{array}{c} \frac{m_{t}-m_{1440}}{m_{0}-m_{1440}},\;while\;water\;desorption\\ \frac{m_{t}-m_{0}}{m_{1440}-m_{0}},\;while\;water\;absorption \end{array}$$
where:$m_{t}$—specimen mass at the time *t* [g];$m_{0}$—specimen mass at the beginning of considered decomposition step [g];$m_{1440}$—sample mass after 1440 min of experiment [g].

### 2.5. Polymer Composite Samples Characterization

#### 2.5.1. Static Mechanical Analysis

Mechanical properties such as tensile strength (TS) and elongation at break (Eb) were determined with the use of a Zwick-Roel Z005 measuring device (Ulm, Germany). Tests were carried out on “dumbbell” shaped specimens that were approximately 1.5 mm thick and 4 mm wide, according to the PN-ISO 37:1998 standard. Specimens were cut out with the use of a punch that was described in the standard. In order to observe the fiber orientation effect, samples for use in the mechanical property determination tests were cut out in the following two directions: vertically (marked as “|”) and horizontally (marked as “−”). Therefore, the cellulose fibers’ orientation influence factor (O) was calculated as follows:(8)$$O=1-\frac{TS_{-}\cdot Eb_{-}}{TS_{|}\cdot Eb_{|}}$$
where:$TS_{-},TS_{|}$—tensile strength of the samples cut out, respectively, vertically and horizontally [MPa];$Eb_{-},Eb_{|}$—elongation at break of the samples cut out, respectively, vertically and horizontally [%].

#### 2.5.2. Dynamic Mechanical Analysis (DMA)

Storage (E′) and loss (E″) moduli were determined with an Ares G2-rheometer that was equipped with parallel plates of 25 mm diameter (stainless steel) from TA Instruments^®^. The soak time was set at 10 s, and the angular frequency was set at 10 rad/s. A logarithmic sweep with a 0.005–70% strain, 20 points per decade, and an initial force of 5 N were chosen. The experiment was carried out in room temperature. Storage and loss moduli values enabled the calculation of the Payne effect (ΔE). The exact equation is given below [59]:(9)$$\Delta E={E^{\prime }}_{max}-{E^{\prime }}_{min}$$
where:${E^{\prime }}_{max},\;{E^{\prime }}_{min}$—storage modulus maximum and minimum value [MPa].

Moreover, in order to assess the cellulose behavior within the polymer matrix, the filler efficiency factor (r) was calculated [60] as follows:(10)$$r=\frac{\frac{{E^{\prime }}_{c}}{{E^{\prime }}_{m}}-1}{V_{F}}$$
where:${E^{\prime }}_{c}$—storage modulus of filled polymer composite [MPa]${E^{\prime }}_{m}$—storage modulus of neat polymer matrix [MPa]$V_{F}$—filler volume fraction [-]

## 3. Results and Discussion

### 3.1. Cellulose Fibers of Various Moisture Contents Influence on Polymer Composite Properties

Here, the moisture content in the cellulose fibers was firstly dried for various times, as has been established with the employment of different techniques that enabled the obtainment of quantitative amounts of water that were adsorbed by the natural filler. Moreover, the processes of water adsorption and desorption have been briefly described. The properties of the cellulose fibers of various water contents were established, and the influence of moisture level in the natural filler on the mechanical and thermal characteristics (Appendix A) of the polymer composites based on an ethylene–norbornene copolymer was investigated.

#### 3.1.1. Cellulose Fibers Characterization

##### FT-IR Investigation

Cellulose fibers were investigated with the employment of spectroscopic methods during the moisture adsorption and desorption. Prior to the adsorption experiment, the biopolymer was dried for 1440 min at 100 °C and then left in ambient conditions for 45, 90, 180 and 1440 min. After the given times were over, the measurements were done immediately.

Moreover, according to the desorption experiment, not-dried cellulose fibers were put into the dryer (100 °C) for the same periods of time as were chosen in the case of the adsorption process analysis. The main goal of the spectroscopic investigation was to observe the water content variations regarding the appropriate absorption band intensities that were assigned to the moisture that was adsorbed by the mentioned biopolymer, e.g., 1650 cm^−1^ (water) [61] and 3331 cm^−1^ (water) [62].

When looking closely at the FT-IR spectra of the cellulose (Figure 2 and Figure 3), one should also notice some characteristic absorption bands, e.g., hydroxyl moieties at 3334 cm^−1^ (–OH) [62] and 1030 cm^−1^ (–OH) [63], C–H stretching vibration [64] at 2896 cm^−1^, –COO [65] at 1200–900 cm^−1^. Absorption bands assigned to the exact moieties has been given in Table 2.

It can be seen in Figure 2 that during the moisture adsorption process, the intensity of the absorption bands at 3331 cm^−1^ and 1653 cm^−1^ increased with the raising time, as was expected from the available literature data [61]. Additionally, the CH stretching bands that were situated at approximately 2900 cm^−1^ were found to be greatly influenced by the moisture content. These fluctuations could be inherent to the material, which is known to be subjected to chemical intrinsic variability or to be assigned to non-specific experimental variation [72]. Furthermore, the changes of water uptake affected the absorption band at 2900 cm^−1^ and the signals at 1030 cm^−1^ (–COO moieties) [65,67] and 557 cm^−1^ (C–OH, C–C bonds) [66].

Similar observations may be done when considering the data shown in Figure 3 that reveal crucial information about water desorption process. Moreover, it should be emphasized that the both moisture adsorption and desorption begin with very high rates that changed in the water content after 45, 90, 180 and 1440 min were hard to be distinguished with the employment of the FT-IR spectra.

##### Fischer Titration and Mass Changes Analysis

Moisture adsorption and desorption processes were tracked with the employment of an analytical weight. Cellulose samples were weighed for a total time of 1440 min during the drying process and moisture adsorption in ambient conditions, as is shown in Figure 4a.

According to the obtained data, it could be claimed that the moisture desorption seemed to be an incredibly fast process, especially at the beginning. Then, an equilibrium state was achieved quite rapidly. In contrast, the water adsorption process seemed to be steadier. Moreover, the plateau that was visible after 400 min of adsorption measurements was only temporary. After 1440 min, the moisture content increased as the sample mass elevated. Similar cellulose sample behavior during both analyzed processes was observed in the case of the NIR analysis. Furthermore, when considering the data that were gathered in Figure 4b, it can be noted that the moisture desorption was a faster process than the water adsorption.

Moreover, according to the results shown in Figure 4b, it can be said that the equilibrium state that is visible in Figure 4a may be not obtained easily. Similarly, variations between the last two curve points that were assigned to the water content level that was obtained with Fischer titration were higher in the case of the moisture adsorption process. Therefore, it may be suggested that the equilibrium state was achieved faster in the case of the water desorption process. Nevertheless, it should be underlined that water adsorption and desorption by cellulose fibers are very complex processes that cannot be described without a further, precise investigation [73,74,75,76].

##### TGA Investigation

According to the thermogravimetric analysis, some differences in the thermal behavior of the analyzed materials were visible. Figure 5 reveals thermogravimetric curves of the UFC100/ND and UFC100/D/1440 samples. Therefore, differences in the water evaporation process and moisture content may be noted. Primary mass loss is always assigned to the volatile matter release (up to 150 °C) [77,78]. The second decomposition step occurs at various temperatures, with the initial point originating from 200–300 °C, and is assigned to cellulose thermal degradation [79].

The most significant differences between two analyzed specimens were visible in the first thermal decomposition step. Then, while the cellulose degraded, both processes started to behave in the same way (Figure 5). A similar trend can also be observed in data shown in Table 3. The most significant differences were detected with an up-to-10% mass loss. Nevertheless, it was T_05%_ that varied the most, shifting its value from 172 °C (UFC100/ND) to 299 °C (UFC100/D/1440). T_05%_ is defined as the temperature at which 5% mass loss was detected. 5% mass loss is more than the water content of cellulose fibers. According to TGA data, the mass loss of the water evaporation step varied between 1.6% and 4.9%.

Furthermore, in order to assess the differences between the modified samples in a more precise way, the activation energies of the decomposition steps were calculated (Table 4). The presented data may be useful tools in describing the water content in the analyzed samples and were in good correspondence with the literature [6]. In general, the lower E_A1_ value, the less moisture was adsorbed by the cellulose fibers [80]. Nevertheless, an elevated water evaporation activation energy of the UFC100/D/1440 sample was observed. This may have been caused by some structural changes that were caused by the thermal treatment and hornifiaction process [81], which relied on carrying out wetting–drying cycles of the biopolymer [48]. Then, some changes in the chemical structure of the natural fiber might have occurred [49,50]. This allowed for a greater dimensional stability and a lower degradation potential (higher molecular packing) [51]. Hornification is commonly carried out in a water environment, and changes in natural fibers could be altered by, e.g., cycle amounts and drying parameters [51,53].

When cellulose fibers are subjected to high temperatures, the internal fiber volume shrinks and the hydrogen bonds are reorganized. Moreover, lactone bridges might be created [82]. If cellulose is then resuspended in water, the original swollen state is not regained [49,50]. The effect of hornification can mostly be observed with properties that are connected to hydration or swelling, e.g., tensile characteristics [83].

#### 3.1.2. Polymer Composite Properties

In further investigation, only not-dried cellulose fibers and the fibers that were dried for 45, 180, and 1440 min (100 °C) were taken into consideration, as there was no significant difference in the moisture content between the specimens that were dried for 45 and 90 min. Figure 6 reveals the digital images of the polymer matrix, the cellulose fibers, and the polymer composite.

##### Static Mechanical Characterization

In order to assess the impact of cellulose moisture content and thermal drying on the mechanical properties of the polymer composites samples, a static mechanical analysis was carried out. In the performed experiment, it could be observed that the tensile strength and elongation at break (Figure 7) rose significantly with the natural filler drying time, as was expected from the literature [84,85,86]. What is surprising is that the drying of the fibers may have caused an increase in the tensile strength from approximately 17 ± 2 MPa (TOPAS and UFC100/ND) to almost 29 ± 5 MPa (TOPAS and UFC100/D/1440). A similar trend was detected in the elongation at break values. Nevertheless, the obtained cellulose-filled polymer composites exhibited a lower performance than the neat polymer matrix: TOPAS − TS = (38 ± 3) MPa; Eb = (490 ± 30)%.

Furthermore, it should be underlined that, as a consequence of the drying process, the impact of cellulose fibers’ orientation within the polymer matrix increases. In the case of the TOPAS and UFC100/ND sample, the material performance did not depend on the direction of the cutting out of the specimens. However, the TOPAS and UC100/D/1440 and TOPAS and UFC100/D/180 samples were characterized by significant differences between the TS| and TS−, as well as the Eb| and Eb−, values. The presented observations are supported by the calculated values of the orientation influence factor (Table 5).

The observed phenomenon may have been caused by differences in filler structure development within a polymer matrix and the anisotropic nature of cellulose fibers, which are high aspect ratio particles [85,87,88]. Nonetheless, it should be underlined that, when considering the mechanical properties, both the direction of the fiber alignment and improvements in dispersion and interfacial adhesion are important factors that contribute to elevated mechanical strength [89,90,91,92].

##### DMA Investigation

Dynamic mechanical analysis is an indispensable tool in rheology properties analysis, e.g., storage (E′) and loss (E″) moduli, the damping factor (tan *δ*), and the Payne effect [93,94,95,96]. This tool provides valuable information about a material that is subjected to dynamically changing forces.

In Figure 8, graphs describing the analyzed composites storage and loss moduli variations during the performed analysis are shown. With an increase of fiber drying time, a more visible Payne effect that is typical for active fillers may be observed—at the point at which E″ achieves its maximum value, E′ starts to drop [97] and the damping factor (tan δ) increases. According to the data available in the literature, the Payne effect may be caused to a faster breakdown of filler-related structures than its recovery [98]. As a consequence, it might help in assessing the filler structure development within a polymer matrix [99]. The raising value of the Payne effect is in a good correspondence with the tensile test results presented in the previous section.

In Table 6, more accurate data considering the Payne effect are given. According to the presented information, the difference between the maximum and minimum values of the storage modulus increased with the cellulose drying time, up to 180 min. However, when the biopolymer was subjected to the thermal treatment (100 °C) for 1440 min, the drop in Payne effect was detected in some cases.

The higher the difference between the maximum and minimum values of the storage modulus is, the more developed a filler structure within a polymer matrix is [100]. Due to this fact, it may be said that cellulose drying helps in building a more advanced filler structure within the TOPAS, and this building is directly connected with an improved filler dispersion and increased cellulose–polymer matrix interactions [100]. Nevertheless, this only occurs to a certain extent because when thermal treatment lasts for too long, some irreversible changes in the natural fiber’s structure occurs, and worse polymer composite properties might be observed.

On the other hand, it can be seen comparing the Payne effect value between the TOPAS and UFC100/D/180 and TOPAS and UFC100/D/1440 samples that the difference was not essential. Therefore, it may be stated that UFC100 could be dried for a shorter time in order to obtain good mechanical properties of its composite. Nevertheless, the drying process is not regular [101,102], and after 1440 min of cellulose thermal treatment, one may be certain that the moisture level decreases significantly in the whole volume of a sample.

Furthermore, according to the data given in Table 7, it may be observed that the cellulose-filled polymer composites’ maximum values of storage and loss moduli were lower than in the case of the neat TOPAS. As a consequence, the presented filler efficiency factor, which compares the storage modulus value while the system is filled and unfilled [60], was, in each case, negative. This proves the well-known fact that cellulose surface modifications caused by certain physical treatments [103,104,105,106] is highly required in order to obtain a material that is characterized by good mechanical properties [107,108,109].

## 4. Conclusions

Based on the gathered data, it may be claimed that the drying of cellulose fibers alters the structure of the biopolymer. This phenomenon was visible the most in this study according to the biopolymer’s thermal properties. The most significant differences were detected as an up-to-10% mass loss—T_05%_ varied the most, shifting its value from 172 °C (UFC100/ND) to 299 °C (UFC100/D/1440).

According to the obtained data, it could be also claimed that the moisture desorption seems to be an incredibly fast process, especially at the beginning. Here, an equilibrium state was rapidly achieved. Contrary, the water adsorption process seems to be steadier, and its changes occur more slowly. Moreover, the plateau visible after 400 min of adsorption measurements was only temporary. After 1440 min, the moisture content increased as the sample mass elevated. However, it was detected that during the drying process, the moisture content in some cases dropped from approximately 6% to almost 1%.

Moving on to the mechanical properties of polymer composite samples, the drying of the fibers resulted in a tensile strength increase from approximately (17 ± 2) MPa (TOPAS and UFC100/ND) to almost (29 ± 5) MPa (TOPAS and UFC100/D/1440)—the performance was almost doubled. A similar trend was evidenced for the elongation at break values. Furthermore, the raising value of the Payne effect corresponded with the presented performance results. Nevertheless, obtained cellulose-filled polymer composites exhibited a lower performance than the neat polymer matrix: TOPAS − TS = (38 ± 3) MPa; Eb = (490 ± 30)%.

However, one must remember that the observed effect was a result, not only of the decreasing moisture content in the natural filler, but also the impact of some structural changes that occurred in the cellulose, e.g., hornification, chemical bridge creation, pore size variations, and hydrogen bond reorganization.

Moreover, cellulose-filled polymer composites may provide an opportunity for the creation of more eco-friendly materials, which are less harmful to the environment, out of commonly used polymers. Cellulose fiber incorporation causes an increase in the degradation potential of prepared materials. Furthermore, due to their high performance, they may find various applications in many industries, such as the packaging and the automotive.

## Figures and Tables

**Figure 1 polymers-12-00484-f001:**
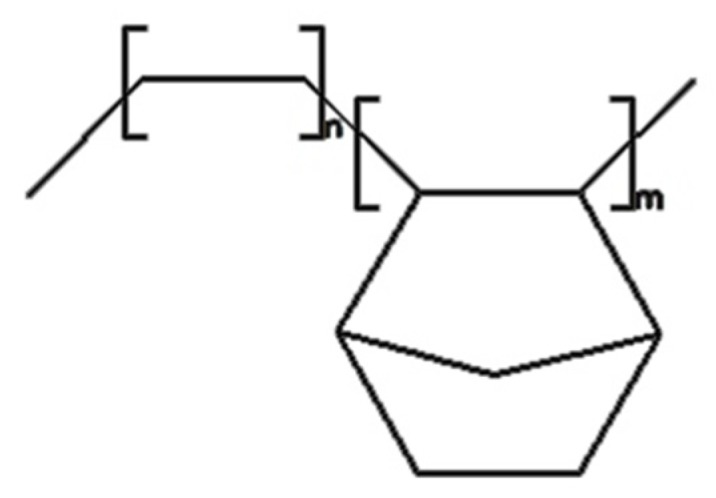
Ethylene–norbornene copolymer structure.

**Figure 2 polymers-12-00484-f002:**
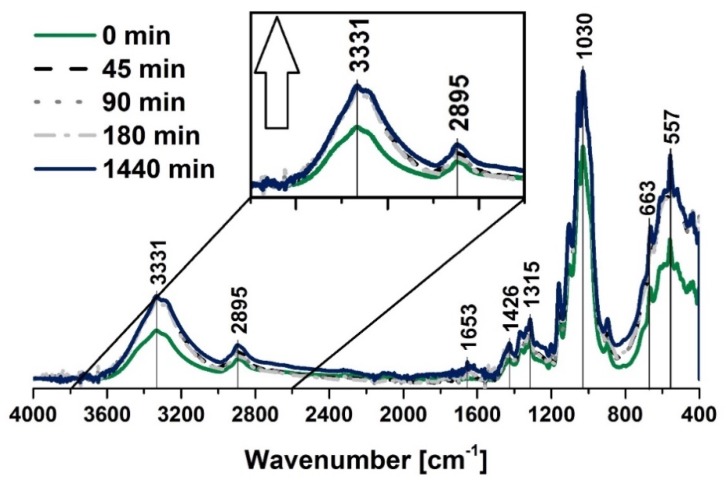
FT-IR spectra of the cellulose fibers that were dried for 1440 min at 100 °C and then left to adsorb moisture for different times. Characteristic absorption bands: 3334 cm^−1^ (O–H, water), 2894 cm^−1^ (C–H), 1200–900 cm^−1^ (O–H, C–O, –COO, CO–O–CO), and 558 cm^−1^ (C–OH, C–C).

**Figure 3 polymers-12-00484-f003:**
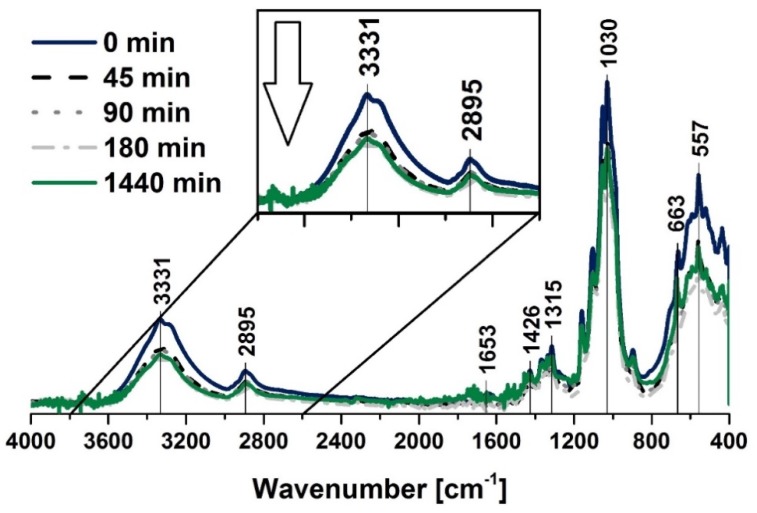
FT-IR spectra of the cellulose fibers that were dried for 45, 90, 180, and 1440 min at 100 °C. Characteristic absorption bands: 3334 cm^−1^ (O–H, water), 2894 cm^−1^ (C–H), 1200–900 cm^−1^ (O–H, C–O, –COO, CO–O–CO), and 558 cm^−1^ (C–OH, C–C).

**Figure 4 polymers-12-00484-f004:**
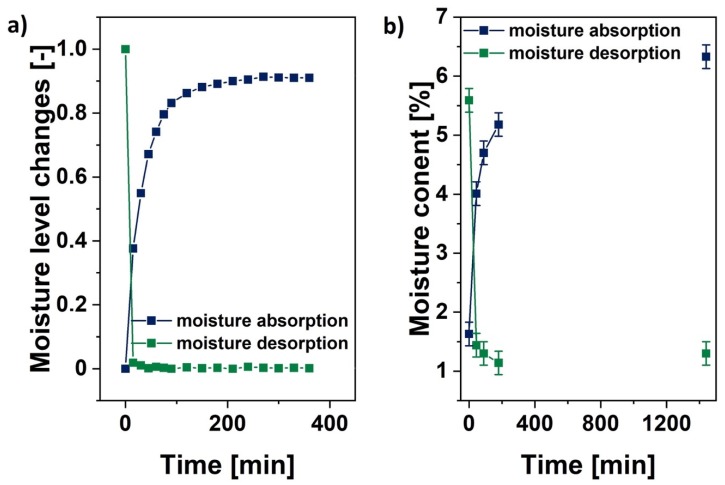
Moisture content changes during the water adsorption and desorption processes. (**a**) Normalized moisture level variations in time according to the sample mass changes and (**b**) Fischer titration experiment results.

**Figure 5 polymers-12-00484-f005:**
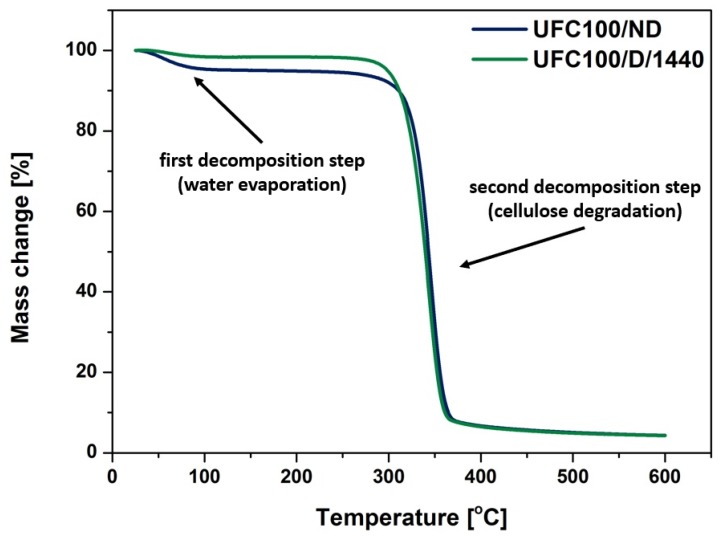
Thermogravimetric analysis (TGA) curves of the dried (1440 min, 100 °C) and not-dried cellulose fibers.

**Figure 6 polymers-12-00484-f006:**
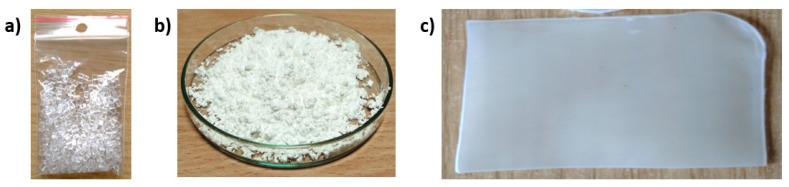
Images of (**a**) the polymer matrix (ethylene–norbornene copolymer), (**b**) the cellulose powder, and (**c**) the polymer composite sample.

**Figure 7 polymers-12-00484-f007:**
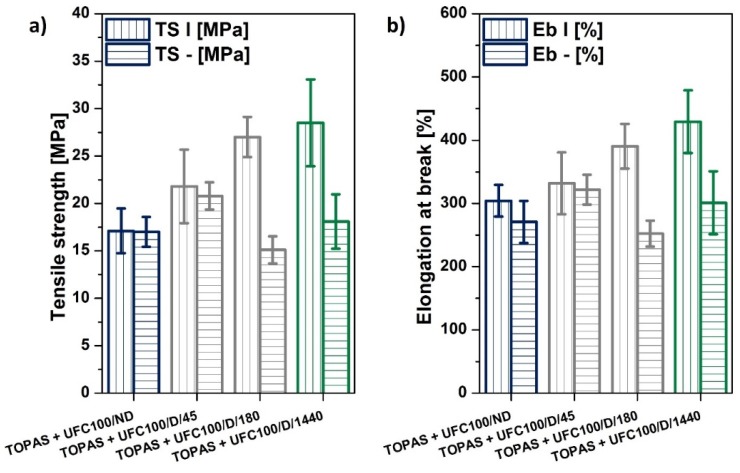
Mechanical properties of the investigated composite samples. (**a**) Tensile strength and (**b**) elongation at break.

**Figure 8 polymers-12-00484-f008:**
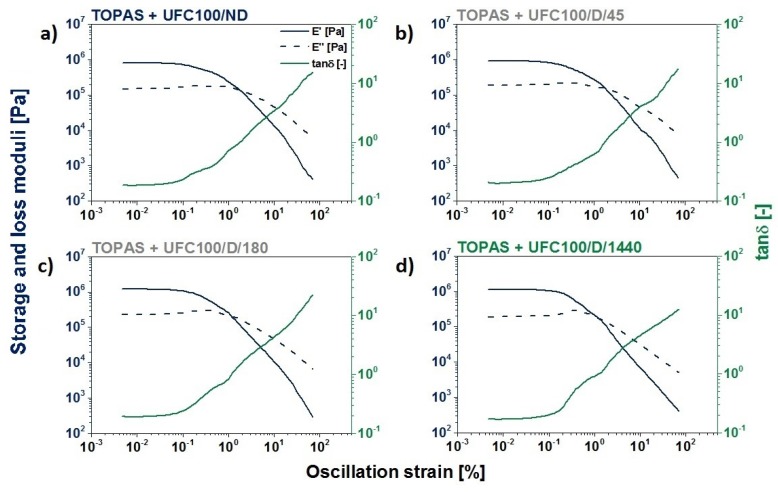
Storage modulus, loss modulus, and tanδ changes of the ethylene–norbornene copolymer (TOPAS) that was filled with cellulose fibers and dried for different times. (**a**) Not-dried, (**b**) dried for 45 min, (**c**) dried for 180 min, (**d**) and dried for 1440 min.

**Table 1 polymers-12-00484-t001:** Drying times of cellulose fibers: D—dried; ND—not-dried.

Sample	Drying Time [min]
UFC100/ND	–
UFC100/D/45	45
UFC100/D/90	90
UFC100/D/180	180
UFC100/D/1440	1440

**Table 2 polymers-12-00484-t002:** Tabularized absorption bands assigned to the chemical groups.

Wavenumber [cm^−1^]	Chemical Group	Ref.
557	C–OH out-of-plane bending, C–C	[66]
1200–900	–OH, –COO	[65]
1100–1000	CO–O–CO	[67]
1058	C–O stretching vibration	[68]
1100	–OH	[63]
1158	C–O stretching vibration, C–O–C bridge	[69]
1245	–CH_3_	[70]
1300–1100	C–O, C=O, C=C, COOH	[63]
1463	C–H bending of CH_2_	[71]
1653	OH bending of adsorbed water, C=C	[61]
2900–2800	CH stretching vibration	[64]
3331	–OH, water	[62]

**Table 3 polymers-12-00484-t003:** Temperatures of the mass loss; T_X%_—temperature at which the mass loss of x% was detected.

Sample	T_05%_ [°C]	T_10%_ [°C]	T_15% %_ [°C]	T_20%_ [°C]	T_50%_ [°C]	T_80%_ [°C]	T_90%_ [°C]
UFC100/ND	172	311	322	328	343	356	365
UFC100/D/1440	299	312	318	323	339	353	361

**Table 4 polymers-12-00484-t004:** Tabularized values of the activation energy that was assigned to the decomposition steps and was calculated with the use of Broido’s method [57]; E_A1_—activation energy of water evaporation process; E_A2_—activation energy of cellulose thermal degradation.

Sample	$E_{A_{1}}\;[\mathrm{kJ}/\mathrm{mol}]$	$E_{A_{2}}\;[\mathrm{kJ}/\mathrm{mol}]$
UFC100/ND	56 ± 2	116 ± 3
UFC100/D/1440	66 ± 2	112 ± 3

**Table 5 polymers-12-00484-t005:** Tabularized values of the orientation influence factor for the investigated composites specimens.

Sample	O [-]
TOPAS and UFC100/ND	0.12
TOPAS and UFC100/D/45	0.07
TOPAS and UFC100/D/180	0.64
TOPAS and UFC100/D/1440	0.55

**Table 6 polymers-12-00484-t006:** Tabularized values of maximum and minimum storage modulus (E′) and the calculated Payne effect.

Sample	E′_max_ [MPa]	E′_min_ [MPa]	Payne Effect [MPa]
TOPAS and UFC100/ND	0.8177	0.0011	0.82
TOPAS and UFC100/D/45	0.9353	0.0014	0.93
TOPAS and UFC100/D/180	1.2062	0.0010	1.21
TOPAS and UFC100/D/1440	1.1396	0.0010	1.14

**Table 7 polymers-12-00484-t007:** Tabularized maximum values of storage (E′), loss moduli (E″), and the calculated filler efficiency factor (r).

Sample	E′_max_ [MPa]	E″_max_ [MPa]	r [-]
TOPAS	1.26	0.32	----------
TOPAS and UFC100/ND	0.82	0.19	−2.5
TOPAS and UFC100/D/45	0.94	0.22	−1.8
TOPAS and UFC100/D/180	1.21	0.30	−0.3
TOPAS and UFC100/D/1440	1.14	0.29	−0.7

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
