# Peer review of "Drying of the Natural Fibers as A Solvent-Free Way to Improve the Cellulose-Filled Polymer Composite Performance"

_polymers, 2020, doi:10.3390/polym12020484_

Round 1

Reviewer 1 Report

The study starts with dried cellulose powder and claims to study the effect of drying. The effect on composite properties can be due to the water content of composite?

The title must be changed/revised.

The paper might have some good science but its really difficult to understand the whole narrative.

Author Response

Institute of Polymer and Dye Technology

Technical University of Lodz

90-924 Lodz, ul Stefanowskiego 12/16, Poland

Tel.: +48 42 631 32 23, Fax: +48 42 636 25 43

February  08, 2020

Polymers

Dear Professor,

We are resubmitting our revised paper entitled Cellulose fibres modification via a hybrid chemical modification by, Stefan Cichosz, Anna Masek with a request to reconsider it for publication in Polymers.

We have carefully considered the Editor and Reviewers' comments. The manuscript was revised exactly according to these comments. The list of responses to the reviewer’s comments and corrections made in the manuscript is attached.

The manuscript has not been previously published, is not currently submitted for review to any other journal, and will not be submitted elsewhere before a decision is made by this journal.

For correspondence please use the following information:

corresponding author: Anna Masek

Institute of Polymer and Dye Technology

Technical University of Lodz

90-924 Lodz, ul Stefanowskiego 12/16, Poland

Tel.: +48 42 631 32 93

Fax: +48 42 636 25 43

Yours sincerely,

Ph. D., D.Sc. Anna Masek

ALL CHANGES IN THE MANUSCRIPT ARE MARKED WITH GREEN COLOUR

Answers to reviewer #1 comments

The comments are listed below.

The study starts with dried cellulose powder and claims to study the effect of drying. The effect on composite properties can be due to the water content of composite?

Answer: As far as we are concerned, there are known many examples of polymer composite properties dependence on the water content, especially in case of hydrophilic materials, e.g.:

Haddad, M. Al Kobaisi, Influence of moisture content on the thermal and mechanical properties and curing behaviour of polymeric matrix and polymer concrete composite, Materials & Design, 2013, 49, 850-856 M. Al-Oqla, S.M. Sapuan, M.R. Ishak, A.A. Nuraini, A novel evaluation tool for enhancing the selection of natural fibers for polymeric composites based on fiber moisture content criterion, BioResources, 2015, 10, 299-312 Kuciel, P. Jakubowska, P. Kuźniar, A study on the mechanical properties and the influence of water uptake and temperature on biocomposites based on polyethylene from renewable sources, Composites Part B: Engineering, 2014, 64, 72-77

Water content is of a huge importance considering the polymer-based composite applications. In general, high water content causes the decrease of both mechanical and thermal properties.

Information given in the Intoduction section: The effect of water amount on polymer composites properties has been detected [54–56]. Yet, it has never been investigated deeply enough. This research provides a reader with a wide range of new and precise data considering the properties of both cellulose and polymer composite regarding different biofiller moisture content (controlled via a drying time).

The title must be changed/revised.

Answer: We are grateful for this comment. The title has been changed for more appropriate one: Drying of the natural fibres as a solvent-free way to improve the cellulose-filled polymer composite performance.

The paper might have some good science but its really difficult to understand the whole narrative.

Answer: We are thankful for Reviewer’s advice. The whole text has been checked once more in order to improve the writing style as well as its readability.

Reviewer 2 Report

The manuscript polymers-710284 studies the influence of the moisture content of cellulose fibers on the properties of polymeric composites.

In my opinion, the manuscript is clearly presented and well written, and I recommend the publication in Polymers journal after major revisions:

Please make a clear differentiation between Abstract, Introduction part and Conclusions. There are the same information, even with the same phrases in all these parts, such as “provides a reader with a wide range of new and precise data considering the properties of both cellulose and polymer composite regarding different biofiller moisture content” (L. 30-23; L. 86-88; L. 514-516)! Moreover, the Abstract contain more words than the accepted limit of Polymers journal, of about 200 words maximum! Please revise these parts and make differences between them, accordingly to the Instructions for Authors! The authors didn’t mention anything about the dispersion of the treated celluloses within matrices and about the compatibility between the matrix and treated UFC100. Please add some information regarding these requests! 3, L. 117: Please explain to what “chemical modification process efficiency” are referred the authors? Regarding the drying process of the samples, I reserve the right to have some doubts regarding the uniformity of the adsorption/desorption phenomena in the experimental system presented by the authors (drying cellulose in powder form in a crystallizer of 70x40 mm)! Please revise the terminology within manuscript! Water absorption test was carried out by immersion samples in water bath, at room temperature for a time duration! The authors studied the adsorption of the cellulose fibers! Can the authors demonstrate the uniformity of the moisture content in the celluloses dried at different times? Moreover, can the authors prove the reproducibility of the obtained results? How significant is the technique to determine the mass variations, given that after drying the samples were left under ambient conditions, with variations of conditions from one determination to another? It is always used a glass desiccator for these experiments! Figure 4 isn’t relevant for the moisture content changes, which take place in the first minutes of the process! Please make them clearer by evidencing the first period of absorption/desorption!

Author Response

Institute of Polymer and Dye Technology

Technical University of Lodz

90-924 Lodz, ul Stefanowskiego 12/16, Poland

Tel.: +48 42 631 32 23, Fax: +48 42 636 25 43

February  08, 2020

Polymers

Dear Professor,

We are resubmitting our revised paper entitled Cellulose fibres modification via a hybrid chemical modification by, Stefan Cichosz, Anna Masek with a request to reconsider it for publication in Polymers.

We have carefully considered the Editor and Reviewers' comments. The manuscript was revised exactly according to these comments. The list of responses to the reviewer’s comments and corrections made in the manuscript is attached.

The manuscript has not been previously published, is not currently submitted for review to any other journal, and will not be submitted elsewhere before a decision is made by this journal.

For correspondence please use the following information:

corresponding author: Anna Masek

Institute of Polymer and Dye Technology

Technical University of Lodz

90-924 Lodz, ul Stefanowskiego 12/16, Poland

Tel.: +48 42 631 32 93

Fax: +48 42 636 25 43

Yours sincerely,

Ph. D., D.Sc. Anna Masek

ALL CHANGES IN THE MANUSCRIPT

Answers to reviewer #2 comments

Reviewer #2: The manuscript polymers-710284 studies the influence of the moisture content of cellulose fibers on the properties of polymeric composites. In my opinion, the manuscript is clearly presented and well written, and I recommend the publication in Polymers journal after major revisions.

The comments are listed below.

Please make a clear differentiation between Abstract, Introduction part and Conclusions. There are the same information, even with the same phrases in all these parts, such as “provides a reader with a wide range of new and precise data considering the properties of both cellulose and polymer composite regarding different biofiller moisture content” (L. 30-23; L. 86-88; L. 514-516)!

Answer: The whole text has revised in accordance with Instruction for Authors section.

Moreover, the Abstract contain more words than the accepted limit of Polymers journal, of about 200 words maximum!

Answer: We are very grateful for drawing our attention to this problem. Abstract part has been shortened and now it consists of less than 200 words: While considering the cellulose (UFC100) modification most of the processes employ various solvents in the role of the reaction environment. Yet, the following article addresses a solvent-free method – thermal drying which causes a moisture content decrease in cellulose fibres. Herein, the moisture content in UFC100 have been analysed with spectroscopic methods, thermogravimetric analysis and differential scanning calorimetry. During water desorption a moisture content drop from approximately 6% to 1% was evidenced. Moreover, drying may bring about a specific variation in the cellulose chemical structure. These changes have affected cellulose filled polymer composite properties, e.g., an increase in tensile strength from 17 MPa for not dried UFC100 to approximately 30 MPa (dried cellulose; 24 h, 100°C) was observed. Furthermore, obtained tensile test results go in a good correspondence with Payne effect values which changes from 0.82 MPa (not dried UFC100) to 1.21 MPa (dried fibres). This raise proves the reinforcing nature of dried UFC100, as Payne effect is dependent on the filler structure development within a polymer matrix. It paves new opportunities for natural fibres application in polymer composites enabling a solvent-free and efficient cellulose modification approach which fulfils the sustainable development rules. (192 words)

The authors didn’t mention anything about the dispersion of the treated celluloses within matrices and about the compatibility between the matrix and treated UFC100. Please add some information regarding these requests!

Answer: We are thankful for the Reviewer’s comment. The dispersion and compatibility between the matrix and UFC100 is examined in the following article in an indirect way. 3.1.2.2. section explains the behaviour of a biofiller within a polymer matrix on the basis of dynamic mechanical analysis – Payne effect and filler efficiency factor have been calculated in order to asses the development of filler structure in the material. Filler structure depends on the dispersion and compatibility as it has been mentioned in the manuscript: The higher the difference between the maximum and minimum value of storage modulus is, the more developed filler structure within a polymer matrix [100]. Due to that fact, it may be said that cellulose drying helps in building more advanced filler structure within TOPAS which is directly connected with an improved filler dispersion and increased cellulose-polymer matrix interactions [100].

 3, L. 117: Please explain to what “chemical modification process efficiency” are referred the authors?

Answer: We are thankful for this comment. It is a mistake and this part has been deleted.

Regarding the drying process of the samples, I reserve the right to have some doubts regarding the uniformity of the adsorption/desorption phenomena in the experimental system presented by the authors (drying cellulose in powder form in a crystallizer of 70x40 mm)! Please revise the terminology within manuscript! Water absorption test was carried out by immersion samples in water bath, at room temperature for a time duration! The authors studied the adsorption of the cellulose fibers! Can the authors demonstrate the uniformity of the moisture content in the celluloses dried at different times?

Answer: We are grateful for this comment. The terminology has been revised. Uniformity of the adsorption/desorption phenomena is a very broad problem which is a subject of  a separate research study (results are not published yet). All of the gathered results presenting the cellulose moisture content are given with the standard deviation values calculated from at least 3 measurements which indicate some information about the uniformity and reproducibility of the results.

Moreover, can the authors prove the reproducibility of the obtained results? How significant is the technique to determine the mass variations, given that after drying the samples were left under ambient conditions, with variations of conditions from one determination to another? It is always used a glass desiccator for these experiments!

Answer: This was an introductory study carried out in order to observe the significance of the changes in polymer composite properties caused by the moisture content in cellulose fibres. As it has been mentioned before, uniformity of the adsorption/desorption phenomena is a very broad problem which is a subject of  a separate research – a follow-up paper. All of the experiments has been carried out under ambient conditions in order to truly mimic the processing process carried out in industry. Furthermore, the absorption experiment has been done over one day, therefore the conditions for all of the samples were similar. We were more interested in the exact value of water amount in cellulose added to the polymer matrix than the process of water adsorption/desorption itself. The main goal of this research study was to investigate the properties of polymer composite filled with cellulose fibres of a different moisture content.

Figure 4 isn’t relevant for the moisture content changes, which take place in the first minutes of the process! Please make them clearer by evidencing the first period of absorption/desorption!

Answer: Fig. 4 has been corrected according to Reviewer #3 comments.

Reviewer 3 Report

Reviewers comments:

The article based on title “Solvent-free way to improve the performance of  cellulose-filled polymer composites for green chemistry” reported by Cichosz et al. is well written and presented. But, still so many quarries need solution for publication.

In introduction, authors can introduce few more references related to your work. Nanomaterials 2019, 9 (11), 1523 and Carbohydrate polymers 2019, 211, 181-194. Some digital images of dispersed cellulose/powder form and film of cellulose/polymer should insert. Check and modify it. Line 102, ……pH value varies between 5-7.5….. needs modification. The presented all equations needs numbering. Line 261-263, no need “thanks” word. should be replaced by Figure. In Fig.4, I think you can time 0-400 min. instead of 0-1220 min. Please check and modify it. The abstract and conclusion part should technical and informative. The abstract should motivate in direction of application at the end of part. Also, conclusion should modify in the same way. 10 needs more quality. Check and modify it. What about standard deviation for tensile strength and elongation at break. Line 505, tensile values needs standard deviation.

Author Response

Institute of Polymer and Dye Technology

Technical University of Lodz

90-924 Lodz, ul Stefanowskiego 12/16, Poland

Tel.: +48 42 631 32 23, Fax: +48 42 636 25 43

February  08, 2020

Polymers

Dear Professor,

We are resubmitting our revised paper entitled Cellulose fibres modification via a hybrid chemical modification by, Stefan Cichosz, Anna Masek with a request to reconsider it for publication in Polymers.

We have carefully considered the Editor and Reviewers' comments. The manuscript was revised exactly according to these comments. The list of responses to the reviewer’s comments and corrections made in the manuscript is attached.

The manuscript has not been previously published, is not currently submitted for review to any other journal, and will not be submitted elsewhere before a decision is made by this journal.

For correspondence please use the following information:

corresponding author: Anna Masek

Institute of Polymer and Dye Technology

Technical University of Lodz

90-924 Lodz, ul Stefanowskiego 12/16, Poland

Tel.: +48 42 631 32 93

Fax: +48 42 636 25 43

Yours sincerely,

Ph. D., D.Sc. Anna Masek

Answers to reviewer #3:

Reviewer #3: The article based on title “Solvent-free way to improve the performance of  cellulose-filled polymer composites for green chemistry” reported by Cichosz et al. is well written and presented. But, still so many quarries need solution for publication.

The comments are listed below. 

In introduction, authors can introduce few more references related to your work. Nanomaterials 2019, 9 (11), 1523 and Carbohydrate polymers 2019, 211, 181-194. Answer: We are very grateful for gathering references that may enrich our article. All of mentioned research studies has been cited. 

Some digital images of dispersed cellulose/powder form and film of cellulose/polymer should insert.

Answer: We are grateful for this advice. Some digital images has been introduced – Fig. 6.

Check and modify it. Line 102, ……pH value varies between 5-7.5….. needs modification.

Answer: As it is mentioned in experimental section, the Arbocel® UFC100 Ultrafine Cellulose for Paper and Board Coating from J. Rettenmaier & Soehne was the type
of cellulose used in this research. It is in a powder form (white, odourless)
and its density is about 1.3 g/cm3. It is insoluble in water and fats. Nevertheless,
this material exhibits a high water binding capacity (even at high temperatures
and shearing forces). Its average fibre length is about 8 μm. pH value varies between
5-7.5. It is a commercially bought cellulose. Information based on the safety data sheet.

The presented all equations needs numbering.

Answer: We are grateful for this comment. The mistake has been corrected. All equations are numbered.

Line 261-263, no need “thanks” word. should be replaced by Figure.

Answer: We could not agree more with this comment. The sentence has been adjusted according to the given comment.

In Fig.4, I think you can time 0-400 min. instead of 0-1220 min. Please check and modify it. modification.

Answer: We thank for drawing our attention to this problem. Fig. 4 has been improved.

The abstract and conclusion part should technical and informative. The abstract should motivate in direction of application at the end of part. Also, conclusion should modify in the same way.

Answer: We are thankful for Reviewer’s advice. The whole text has revised in accordance with Instruction for Authors section.

10 needs more quality. Check and modify it.

Answer: Fig. 10 does not exist in this document. Nevertheless, we have checked all of the figures considering their quality and data readability. Some of them has been improved.

What about standard deviation for tensile strength and elongation at break. Line 505, tensile values needs standard deviation.

Answer: We are terribly sorry for this oversight. The mistake has been corrected.

Reviewer 4 Report

Dear author

The paper is interesting and it contains valuable data for the field of polymer composite materials. Some comments:

Novelty of the paper: it is not clear which are the novelty elements in the paper with respect to literature Why did you choose ethylene-norbornene copolymer as the polymer matrix? Pay attention to English errors: biopolimer instead of biopolymer for i.e. You shoudl check again the manuscript. Provide stress-strain curves for your samples (in the revision report not in the manuscript) What about the morphology of the prepared composites (SEM, AFM)? Did you perform XRD analysis? This could be interesting with respect to cellulose crystallinity changes.

Author Response

Institute of Polymer and Dye Technology

Technical University of Lodz

90-924 Lodz, ul Stefanowskiego 12/16, Poland

Tel.: +48 42 631 32 23, Fax: +48 42 636 25 43

February  08, 2020

Polymers

Dear Professor,

We are resubmitting our revised paper entitled Cellulose fibres modification via a hybrid chemical modification by, Stefan Cichosz, Anna Masek with a request to reconsider it for publication in Polymers.

We have carefully considered the Editor and Reviewers' comments. The manuscript was revised exactly according to these comments. The list of responses to the reviewer’s comments and corrections made in the manuscript is attached.

The manuscript has not been previously published, is not currently submitted for review to any other journal, and will not be submitted elsewhere before a decision is made by this journal.

For correspondence please use the following information:

corresponding author: Anna Masek

Institute of Polymer and Dye Technology

Technical University of Lodz

90-924 Lodz, ul Stefanowskiego 12/16, Poland

Tel.: +48 42 631 32 93

Fax: +48 42 636 25 43

Yours sincerely,

Ph. D., D.Sc. Anna Masek

Answers to reviewer #4:

Reviewer #4: The paper is interesting and it contains valuable data for the field of polymer composite materials.

The comments are listed below.

Novelty of the paper: it is not clear which are the novelty elements in the paper with respect to literature.

Answer: We are grateful for drawing our attention to this problem. The novelty has been described more precisely: The effect of water amount on polymer composites properties has been detected [54–56]. Yet, it has never been investigated deeply enough. Different properties origin has not been explained. This research provides a reader with a wide range of new and precise data considering the properties of both cellulose and polymer composite regarding different biofiller moisture content (controlled via a drying time). On the basis of gathered information, it could be observed that with a simple drying process of cellulose, the polymer composite performance may be doubled. Given article is an introduction to a research analysis on the consequences of water content amount in cellulose on the properties of natural fibres loaded polymer composite properties, as well as biopolymer structure and features.

Why did you choose ethylene-norbornene copolymer as the polymer matrix?

Answer: This is a thermoplastic elastomer of a very interesting properties and easy processing procedure. This material is a high-performance alternative to traditional flexible materials for use in a broad range of applications, such as medical devices, injection molding articles for optical industry, packaging Industry. It was also a subject of our previous research studies:

Cichosz, A. Masek, K. Wolski, Innovative cellulose fibres reinforced ethylene-norbornene copolymer composites of an increased degradation potential, Polymer Degradation and Stability, 2019, 159, 174-183. doi: 10.1016/j.polymdegradstab.2018.11.029 Wolski, S. Cichosz, A. Masek, Surface hydrophobisation of lignocellulosic waste for the preparation of biothermoelastoplastic composites, European Polymer Journal, 2019, 118, 481-491. doi: 10.1016/j.eurpolymj.2019.06.026

Moreover, the overriding aim of the research is to incorporate biocomponents into commonly used polymer matrices – increase of material degradation potential and creation of a polymer composite less harmful to the natural environment. It has been mentioned in Conclusions section: Moreover, cellulose-filled polymer composites may bring an opportunity for creation of more eco-friendly materials, which are less harmful for the environment, out of commonly used polymers. Cellulose fibres incorporation causes an increase in a degradation potential of prepared materials. Furthermore, due to the high performance, they may find various applications in many branches of industry, from packaging to automotive. 

Pay attention to English errors: biopolimer instead of biopolymer for i.e. You should check again the manuscript.

Answer: We are thankful for Reviewer’s comment. Manuscript has been revised again considering the language style, mistakes and readability.

Provide stress-strain curves for your samples (in the revision report not in the manuscript).

Answer: Stress-strain curves are provided below:

TOPAS + UFC100/ND –

TOPAS + UFC100/ND |

TOPAS + UFC100/D/45 -

TOPAS + UFC100/D/45 |

TOPAS + UFC100/D/180 -

TOPAS + UFC100/D/180 |

TOPAS + UFC100/D/1440 |

TOPAS + UFC100/D/1440 -

What about the morphology of the prepared composites (SEM, AFM)? Did you perform XRD analysis? This could be interesting with respect to cellulose crystallinity changes.

Answer: We are very grateful for this comment and idea for extended research. It is incredibly valuable. Unfortunately, we are not able to carry out such experiments right now. We plan to employ XRD and SEM in the further research, as this was an introductory study carried out in order to observe the significance of the changes in polymer composite properties caused by the moisture content in cellulose fibres. The problem of moisture content in cellulose fibres and its impact on cellulose-filled polymer composites is going to be investigated more deeply in the future.

Answers to reviewer #4:

Reviewer #4: The paper is interesting and it contains valuable data for the field of polymer composite materials.

The comments are listed below.

Novelty of the paper: it is not clear which are the novelty elements in the paper with respect to literature.

Answer: We are grateful for drawing our attention to this problem. The novelty has been described more precisely: The effect of water amount on polymer composites properties has been detected [54–56]. Yet, it has never been investigated deeply enough. Different properties origin has not been explained. This research provides a reader with a wide range of new and precise data considering the properties of both cellulose and polymer composite regarding different biofiller moisture content (controlled via a drying time). On the basis of gathered information, it could be observed that with a simple drying process of cellulose, the polymer composite performance may be doubled. Given article is an introduction to a research analysis on the consequences of water content amount in cellulose on the properties of natural fibres loaded polymer composite properties, as well as biopolymer structure and features.

Why did you choose ethylene-norbornene copolymer as the polymer matrix?

Answer: This is a thermoplastic elastomer of a very interesting properties and easy processing procedure. This material is a high-performance alternative to traditional flexible materials for use in a broad range of applications, such as medical devices, injection molding articles for optical industry, packaging Industry. It was also a subject of our previous research studies:

Cichosz, A. Masek, K. Wolski, Innovative cellulose fibres reinforced ethylene-norbornene copolymer composites of an increased degradation potential, Polymer Degradation and Stability, 2019, 159, 174-183. doi: 10.1016/j.polymdegradstab.2018.11.029 Wolski, S. Cichosz, A. Masek, Surface hydrophobisation of lignocellulosic waste for the preparation of biothermoelastoplastic composites, European Polymer Journal, 2019, 118, 481-491. doi: 10.1016/j.eurpolymj.2019.06.026

Moreover, the overriding aim of the research is to incorporate biocomponents into commonly used polymer matrices – increase of material degradation potential and creation of a polymer composite less harmful to the natural environment. It has been mentioned in Conclusions section: Moreover, cellulose-filled polymer composites may bring an opportunity for creation of more eco-friendly materials, which are less harmful for the environment, out of commonly used polymers. Cellulose fibres incorporation causes an increase in a degradation potential of prepared materials. Furthermore, due to the high performance, they may find various applications in many branches of industry, from packaging to automotive. 

Pay attention to English errors: biopolimer instead of biopolymer for i.e. You should check again the manuscript.

Answer: We are thankful for Reviewer’s comment. Manuscript has been revised again considering the language style, mistakes and readability.

Provide stress-strain curves for your samples (in the revision report not in the manuscript).

Answer: Stress-strain curves are provided below:

TOPAS + UFC100/ND –

TOPAS + UFC100/ND |

TOPAS + UFC100/D/45 -

TOPAS + UFC100/D/45 |

TOPAS + UFC100/D/180 -

TOPAS + UFC100/D/180 |

TOPAS + UFC100/D/1440 |

TOPAS + UFC100/D/1440 -

What about the morphology of the prepared composites (SEM, AFM)? Did you perform XRD analysis? This could be interesting with respect to cellulose crystallinity changes.

Answer: We are very grateful for this comment and idea for extended research. It is incredibly valuable. Unfortunately, we are not able to carry out such experiments right now. We plan to employ XRD and SEM in the further research, as this was an introductory study carried out in order to observe the significance of the changes in polymer composite properties caused by the moisture content in cellulose fibres. The problem of moisture content in cellulose fibres and its impact on cellulose-filled polymer composites is going to be investigated more deeply in the future.

Answers to reviewer #4:

Reviewer #4: The paper is interesting and it contains valuable data for the field of polymer composite materials.

The comments are listed below.

Novelty of the paper: it is not clear which are the novelty elements in the paper with respect to literature.

Answer: We are grateful for drawing our attention to this problem. The novelty has been described more precisely: The effect of water amount on polymer composites properties has been detected [54–56]. Yet, it has never been investigated deeply enough. Different properties origin has not been explained. This research provides a reader with a wide range of new and precise data considering the properties of both cellulose and polymer composite regarding different biofiller moisture content (controlled via a drying time). On the basis of gathered information, it could be observed that with a simple drying process of cellulose, the polymer composite performance may be doubled. Given article is an introduction to a research analysis on the consequences of water content amount in cellulose on the properties of natural fibres loaded polymer composite properties, as well as biopolymer structure and features.

Why did you choose ethylene-norbornene copolymer as the polymer matrix?

Answer: This is a thermoplastic elastomer of a very interesting properties and easy processing procedure. This material is a high-performance alternative to traditional flexible materials for use in a broad range of applications, such as medical devices, injection molding articles for optical industry, packaging Industry. It was also a subject of our previous research studies:

Cichosz, A. Masek, K. Wolski, Innovative cellulose fibres reinforced ethylene-norbornene copolymer composites of an increased degradation potential, Polymer Degradation and Stability, 2019, 159, 174-183. doi: 10.1016/j.polymdegradstab.2018.11.029 Wolski, S. Cichosz, A. Masek, Surface hydrophobisation of lignocellulosic waste for the preparation of biothermoelastoplastic composites, European Polymer Journal, 2019, 118, 481-491. doi: 10.1016/j.eurpolymj.2019.06.026

Moreover, the overriding aim of the research is to incorporate biocomponents into commonly used polymer matrices – increase of material degradation potential and creation of a polymer composite less harmful to the natural environment. It has been mentioned in Conclusions section: Moreover, cellulose-filled polymer composites may bring an opportunity for creation of more eco-friendly materials, which are less harmful for the environment, out of commonly used polymers. Cellulose fibres incorporation causes an increase in a degradation potential of prepared materials. Furthermore, due to the high performance, they may find various applications in many branches of industry, from packaging to automotive.

Pay attention to English errors: biopolimer instead of biopolymer for i.e. You should check again the manuscript.

Answer: We are thankful for Reviewer’s comment. Manuscript has been revised again considering the language style, mistakes and readability.

Provide stress-strain curves for your samples (in the revision report not in the manuscript).

Answer: Stress-strain curves are provided below:

TOPAS + UFC100/ND –

TOPAS + UFC100/ND |

TOPAS + UFC100/D/45 -

TOPAS + UFC100/D/45 |

TOPAS + UFC100/D/180 -

TOPAS + UFC100/D/180 |

TOPAS + UFC100/D/1440 |

TOPAS + UFC100/D/1440 -

What about the morphology of the prepared composites (SEM, AFM)? Did you perform XRD analysis? This could be interesting with respect to cellulose crystallinity changes.

Answer: We are very grateful for this comment and idea for extended research. It is incredibly valuable. Unfortunately, we are not able to carry out such experiments right now. We plan to employ XRD and SEM in the further research, as this was an introductory study carried out in order to observe the significance of the changes in polymer composite properties caused by the moisture content in cellulose fibres. The problem of moisture content in cellulose fibres and its impact on cellulose-filled polymer composites is going to be investigated more deeply in the future.

Answers to reviewer #4:

Reviewer #4: The paper is interesting and it contains valuable data for the field of polymer composite materials.

The comments are listed below.

Novelty of the paper: it is not clear which are the novelty elements in the paper with respect to literature.

Answer: We are grateful for drawing our attention to this problem. The novelty has been described more precisely: The effect of water amount on polymer composites properties has been detected [54–56]. Yet, it has never been investigated deeply enough. Different properties origin has not been explained. This research provides a reader with a wide range of new and precise data considering the properties of both cellulose and polymer composite regarding different biofiller moisture content (controlled via a drying time). On the basis of gathered information, it could be observed that with a simple drying process of cellulose, the polymer composite performance may be doubled. Given article is an introduction to a research analysis on the consequences of water content amount in cellulose on the properties of natural fibres loaded polymer composite properties, as well as biopolymer structure and features.

Why did you choose ethylene-norbornene copolymer as the polymer matrix?

Answer: This is a thermoplastic elastomer of a very interesting properties and easy processing procedure. This material is a high-performance alternative to traditional flexible materials for use in a broad range of applications, such as medical devices, injection molding articles for optical industry, packaging Industry. It was also a subject of our previous research studies:

Cichosz, A. Masek, K. Wolski, Innovative cellulose fibres reinforced ethylene-norbornene copolymer composites of an increased degradation potential, Polymer Degradation and Stability, 2019, 159, 174-183. doi: 10.1016/j.polymdegradstab.2018.11.029 Wolski, S. Cichosz, A. Masek, Surface hydrophobisation of lignocellulosic waste for the preparation of biothermoelastoplastic composites, European Polymer Journal, 2019, 118, 481-491. doi: 10.1016/j.eurpolymj.2019.06.026

Moreover, the overriding aim of the research is to incorporate biocomponents into commonly used polymer matrices – increase of material degradation potential and creation of a polymer composite less harmful to the natural environment. It has been mentioned in Conclusions section: Moreover, cellulose-filled polymer composites may bring an opportunity for creation of more eco-friendly materials, which are less harmful for the environment, out of commonly used polymers. Cellulose fibres incorporation causes an increase in a degradation potential of prepared materials. Furthermore, due to the high performance, they may find various applications in many branches of industry, from packaging to automotive.

Pay attention to English errors: biopolimer instead of biopolymer for i.e. You should check again the manuscript.

Answer: We are thankful for Reviewer’s comment. Manuscript has been revised again considering the language style, mistakes and readability.

Provide stress-strain curves for your samples (in the revision report not in the manuscript).

Answer: Stress-strain curves are provided below:

TOPAS + UFC100/ND –

TOPAS + UFC100/ND |

TOPAS + UFC100/D/45 -

TOPAS + UFC100/D/45 |

TOPAS + UFC100/D/180 -

TOPAS + UFC100/D/180 |

TOPAS + UFC100/D/1440 |

TOPAS + UFC100/D/1440 -

What about the morphology of the prepared composites (SEM, AFM)? Did you perform XRD analysis? This could be interesting with respect to cellulose crystallinity changes.

Answer: We are very grateful for this comment and idea for extended research. It is incredibly valuable. Unfortunately, we are not able to carry out such experiments right now. We plan to employ XRD and SEM in the further research, as this was an introductory study carried out in order to observe the significance of the changes in polymer composite properties caused by the moisture content in cellulose fibres. The problem of moisture content in cellulose fibres and its impact on cellulose-filled polymer composites is going to be investigated more deeply in the future.

Answers to reviewer #4:

Reviewer #4: The paper is interesting and it contains valuable data for the field of polymer composite materials.

The comments are listed below.

Novelty of the paper: it is not clear which are the novelty elements in the paper with respect to literature.

Answer: We are grateful for drawing our attention to this problem. The novelty has been described more precisely: The effect of water amount on polymer composites properties has been detected [54–56]. Yet, it has never been investigated deeply enough. Different properties origin has not been explained. This research provides a reader with a wide range of new and precise data considering the properties of both cellulose and polymer composite regarding different biofiller moisture content (controlled via a drying time). On the basis of gathered information, it could be observed that with a simple drying process of cellulose, the polymer composite performance may be doubled. Given article is an introduction to a research analysis on the consequences of water content amount in cellulose on the properties of natural fibres loaded polymer composite properties, as well as biopolymer structure and features.

Why did you choose ethylene-norbornene copolymer as the polymer matrix?

Answer: This is a thermoplastic elastomer of a very interesting properties and easy processing procedure. This material is a high-performance alternative to traditional flexible materials for use in a broad range of applications, such as medical devices, injection molding articles for optical industry, packaging Industry. It was also a subject of our previous research studies:

Cichosz, A. Masek, K. Wolski, Innovative cellulose fibres reinforced ethylene-norbornene copolymer composites of an increased degradation potential, Polymer Degradation and Stability, 2019, 159, 174-183. doi: 10.1016/j.polymdegradstab.2018.11.029 Wolski, S. Cichosz, A. Masek, Surface hydrophobisation of lignocellulosic waste for the preparation of biothermoelastoplastic composites, European Polymer Journal, 2019, 118, 481-491. doi: 10.1016/j.eurpolymj.2019.06.026

Moreover, the overriding aim of the research is to incorporate biocomponents into commonly used polymer matrices – increase of material degradation potential and creation of a polymer composite less harmful to the natural environment. It has been mentioned in Conclusions section: Moreover, cellulose-filled polymer composites may bring an opportunity for creation of more eco-friendly materials, which are less harmful for the environment, out of commonly used polymers. Cellulose fibres incorporation causes an increase in a degradation potential of prepared materials. Furthermore, due to the high performance, they may find various applications in many branches of industry, from packaging to automotive.

Pay attention to English errors: biopolimer instead of biopolymer for i.e. You should check again the manuscript.

Answer: We are thankful for Reviewer’s comment. Manuscript has been revised again considering the language style, mistakes and readability.

Provide stress-strain curves for your samples (in the revision report not in the manuscript).

Answer: Stress-strain curves are provided below:

TOPAS + UFC100/ND –

TOPAS + UFC100/ND |

TOPAS + UFC100/D/45 -

TOPAS + UFC100/D/45 |

TOPAS + UFC100/D/180 -

TOPAS + UFC100/D/180 |

TOPAS + UFC100/D/1440 |

TOPAS + UFC100/D/1440 -

What about the morphology of the prepared composites (SEM, AFM)? Did you perform XRD analysis? This could be interesting with respect to cellulose crystallinity changes.

Answer: We are very grateful for this comment and idea for extended research. It is incredibly valuable. Unfortunately, we are not able to carry out such experiments right now. We plan to employ XRD and SEM in the further research, as this was an introductory study carried out in order to observe the significance of the changes in polymer composite properties caused by the moisture content in cellulose fibres. The problem of moisture content in cellulose fibres and its impact on cellulose-filled polymer composites is going to be investigated more deeply in the future.

Round 2

Reviewer 1 Report

The authors have addressed reviwer's comments. I reccommend to accept it

Reviewer 2 Report

In my opinion, the authors paid attention to the comments, have made improvements to the manuscript polymers-710284, as compare to previous version, and thus, I recommend the publication of the present version in Polymers journal.

Reviewer 3 Report

Accept in current form.